# Specimen Geometry Effect on Experimental Tensile Mechanical Properties of Tough Hydrogels

**DOI:** 10.3390/ma16020785

**Published:** 2023-01-13

**Authors:** Donghwan Ji, Pilseon Im, Sunmi Shin, Jaeyun Kim

**Affiliations:** 1School of Chemical Engineering, Sungkyunkwan University (SKKU), Suwon 16419, Republic of Korea; 2Department of Mechanical Engineering, National University of Singapore (NUS), Singapore 117575, Singapore; 3Department of Health Sciences and Technology, Samsung Advanced Institute for Health Sciences & Technology (SAIHST), Sungkyunkwan University (SKKU), Suwon 16419, Republic of Korea; 4Biomedical Institute for Convergence at SKKU (BICS), Sungkyunkwan University (SKKU), Suwon 16419, Republic of Korea; 5Institute of Quantum Biophysics (IQB), Sungkyunkwan University (SKKU), Suwon 16419, Republic of Korea

**Keywords:** double-network hydrogel, tough hydrogel, tensile testing, mechanical properties

## Abstract

Synthetic tough hydrogels have received attention because they could mimic the mechanical properties of natural hydrogels, such as muscle, ligament, tendon, and cartilage. Many recent studies suggest various approaches to enhance the mechanical properties of tough hydrogels. However, directly comparing each hydrogel property in different reports is challenging because various testing specimen shapes/sizes were employed, affecting the experimental mechanical property values. This study demonstrates how the specimen geometry—the lengths and width of the reduced section—of a tough double-network hydrogel causes differences in experimental tensile mechanical values. In particular, the elastic modulus was systemically compared using eleven specimens of different shapes and sizes that were tensile tested, including a rectangle, ASTM D412-C and D412-D, JIS K6251-7, and seven customized dumbbell shapes with various lengths and widths of the reduced section. Unlike the rectangular specimen, which showed an inconsistent measurement of mechanical properties due to a local load concentration near the grip, dumbbell-shaped specimens exhibited a stable fracture at the reduced section. The dumbbell-shaped specimen with a shorter gauge length resulted in a smaller elastic modulus. Moreover, a relationship between the specimen dimension and measured elastic modulus value was derived, which allowed for the prediction of the experimental elastic modulus of dumbbell-shaped tough hydrogels with different dimensions. This study conveys a message that reminds the apparent experimental dependence of specimen geometry on the stress-strain measurement and the need to standardize the measurement of of numerous tough hydrogels for a fair comparison.

## 1. Introduction

Hydrogels are a significant component of the human body, consisting primarily of three-dimensional polymer networks and water. Natural hydrogels, such as muscle, ligament, tendon, cartilage, and so on, commonly suffer from mechanical damage due to repetitive exercise of tension and compression. For that reason, soft yet tough synthetic hydrogels, including double-network (DN) and polyampholyte (PA) hydrogels [1,2,3,4], are emerging to support/replace biological tissues (natural hydrogels) and to further bridge human–machine interfaces as adhesives and surface coating materials. Since a mechanical match between the synthetic hydrogel and surrounding tissues is one of the primary considerations in the development of the practical use of synthetic hydrogels [5,6,7,8,9], many researchers have focused on improving the hydrogel mechanical performance.

The hydrogel mechanical properties are usually evaluated through tensile testing using a universal testing machine (UTM) that uses values such as the yield strength, fracture strength, elastic modulus, work of fracture, and fracture toughness. However, because previous reports indicate that hydrogels have a wide variety of specimen shapes and sizes, the improvement of hydrogel mechanical properties compared with the values in previous reports is unclear. Despite this, studies regarding the standardization of hydrogel specimens for a fair comparison of their mechanical properties are lacking.

In this study, considering the hypothesis that the mechanical property values vary with the test specimen shapes/sizes, we used tensile testing to investigate the influence of the specimen shapes and sizes of a tough hydrogel on the experimental tensile mechanical properties. To systemically compare the values, we prepared variously shaped alginate/polyacrylamide (Alg/PAM) double-network hydrogels as tough-hydrogel models. We found that the rectangular specimen is inappropriate for measuring accurate mechanical properties owing to the stress concentration near the grip, whereas the dumbbell-shaped specimen is more suitable for obtaining consistent mechanical property values owing to stable stretching up to the occurrence of a fracture at a reduced section rather than near the grip. Furthermore, we derived a relationship between the specimen dimensions and measured elastic modulus values using dumbbell-shaped specimens, allowing for the prediction of the elastic moduli based on the hydrogel specimen dimensions.

## 2. Experimental Section

### 2.1. Materials

Alginic acid sodium salt from brown algae (Alg, medium viscosity, Sigma A2033, Sigma-Aldrich, St. Lois, MO, USA), acrylamide (AM, Sigma A8887), N,N′-methylenebisacrylamide (MBAA, Sigma M7279), N,N,N′,N′-tetramethylethylenediamine (TMEDA, Sigma T7024), ammonium persulfate (APS, Sigma A7460), and calcium sulfate dihydrate (CaSO_4_·2H_2_O, Samchun C0227, Samchun Chemical Co., Ltd., Seoul, Republic of Korea) were purchased and used without any purification.

### 2.2. Preparation of Alginate/Polyacrylamide (Alg/PAM) Double-Network (DN) Tough Hydrogels

The Alg and AM were first dissolved in distilled water. Subsequently, MBAA (0.07% *wt*/*wt* of AM), a covalent crosslinker for AM; TMEDA (0.28% *wt*/*wt* of AM), an accelerator for AM; APS (4% *wt*/*wt* of AM), an initiator for AM; and CaSO_4_ (15% *wt*/*wt* of Alg), an ionic crosslinker for Alg were added to the Alg/AM solution to obtain a 1.83 wt% Alg and 12 wt% AM mixture. The obtained mixture was poured into a rectangular mold with a thickness of 3 mm and polymerized under 254 nm ultraviolet light for an hour. The resulting double network (DN) hydrogel composed of Ca^2+^-crosslinked alginate and covalently crosslinked polyacrylamide (Alg/PAM) hydrogel was then stored overnight in a cooler maintained at approximately 5 °C. In the case of Fe-crosslinked DN hydrogels, the Alg/PAM hydrogel was soaked in 100 mM FeCl_3_ solution for one day.

### 2.3. Preparation of Hydrogel Specimens

Eleven specimens of different shapes and sizes were prepared, including a rectangle, ASTM-standard ASTM D412-C and D412-D, Japan-standard JIS K6251-7, and seven customized dumbbell shapes (#1–#7) with various lengths and widths of the reduced section, by cutting a large and wide hydrogel into the specific geometry. The detailed specimen shapes and sizes are depicted in the main figures.

### 2.4. Mechanical Test

Tensile testing was conducted using a Comtech QC548 M1F-M (Comtech, Testing Machines Co., Ltd., Taichung City, Taiwan) universal testing machine with a 100 N load cell. The tests were performed at a load speed of 100 mm min^−1^. The elastic modulus was measured using the slope of the linear elastic region.

### 2.5. Statistical Analysis

The mechanical testing of each hydrogel was performed at least five times, and the consequent data were presented as the average ± standard deviation. Statistical significance was conducted using Student’s *t*-test (one-sided test); ** *p* < 0.01; *** *p* < 0.001; **** *p* < 0.0001; n.s., not significant.

## 3. Results and Discussion

### 3.1. Difference between Rectangle and ASTM and JIS Standards

The double-network (DN) hydrogels, which are composed of two interpenetrating polymer networks, a rigid first-network and a stretchable second-network, with contrasting mechanical properties, have a significant ability to absorb mechanical energy that is incomparable to single-network (SN) hydrogels [9,10]. Among many kinds of DN tough hydrogels, the Alg/PAM hydrogel, consisting of physically Ca-crosslinked Alg and chemically covalent-crosslinked PAM, is a representatively widely used DN hydrogel due to the easy one-pot synthesis method [2]. The Alg, a natural polysaccharide polymer, which has G residues (G blocks), can be coordinated with metal cations, forming rigid polymer networks. These relatively rigid Alg networks can be combined with relatively stretchable PAM networks, becoming tough DN hydrogels. Based on the same principle and fabrication procedure as for the Alg/PAM DN hydrogel, other kinds of DN tough hydrogels that are composed of a different first-network polymer, such as agar, chitosan, cellulose, polyvinyl alcohol (PVA), i.e., Agar/PAM [11,12], Chitosan/PAM [13,14,15,16], Cellulose/PAM [17], and PVA/PAM [18] DN hydrogels, were also proposed. In addition, post-treatments to enhance the mechanical properties of the Alg/PAM DN hydrogel were usually also demonstrated [6,19,20,21,22,23,24,25]. Meanwhile, a wide variety of specimen geometries (shapes and sizes) used in each research paper hinder a direct comparison of how much mechanical improvement was achieved compared to the previous hydrogels (Table 1). Accordingly, herein, we would like to figure out to what extent the hydrogel specimen geometry affects the tensile mechanical testing results, mainly, an elastic modulus value that is one of the most important factors in judging the material’s usability. 

First, we prepared four types of hydrogels with frequently used standardized shapes: rectangular, ASTM standard ASTM D412-C and D412-D, and Japan-standard JIS K6251-7 (Figure 1a). The rectangular shape is the most frequently used shape to measure the tensile mechanical properties of hydrogels, including tough gels, owing to the ease of specimen preparation. In the case of a dumbbell shape, the ASTM D412-C is one of the most frequently used dumbbell shapes to measure stretchable materials such as rubbers and elastomers, and it has a width of 6 mm and a gauge length of 33 mm in a reduced section [26]. ASTM D412-D is also typically used to measure such stretchable materials, with a width of 3 mm and a gauge length of 33 mm in a reduced section [26]. JIS K6251-7 has one of the smallest sizes, and its width and gauge length in a reduced section are 2 and 12 mm, respectively [27].

Since hydrogels are soft and weak, measuring rectangular hydrogels in tensile tests requires special care. When the hydrogel was weakly clamped, causing the hydrogel to slip from the grip, the stress-strain curves were not accurate (Figure 1b, dotted line). Some cases exhibited irregular stress values at large strain regions because of gradual slipping from the grip, whereas others demonstrated low final stress and strain values because the hydrogel abruptly came out of the grip early. Such challenges consequently led to large deviations in the elastic modulus values. On the other hand, when the hydrogel was adequately clamped without any scope for it to slip from the grips, the stress-strain curves and elastic moduli were evenly obtained (Figure 1b, solid line). However, in many cases, a stress concentration near the grip caused an early fracture of the hydrogel (Figure 1c), obscuring an accurate evaluation of the hydrogel fracture strength and strain. In most reports on tough hydrogels, rectangular specimens were used; therefore, the assessment of the mechanical properties of newly developed hydrogels by comparing them with those of previously reported hydrogels can be limited.

In contrast to the rectangular hydrogel, all dumbbell-shaped Alg/PAM tough hydrogels, which contained a reduced section where almost all of the load was gathered, exhibited more consistent stress-strain curves with a large strain until fracture occurred at the reduced section in different samples with the same shape when compared with that of the rectangular shape (Figure 1d). Since the stress was concentrated in the narrower region rather than in the wider clamping region, the fracture was consistently observed in the reduced section. Therefore, dumbbell-shaped hydrogels are more suitable than rectangular hydrogels for accurately and consistently evaluating hydrogel tensile properties. However, a notable issue exists in the result for dumbbell-shaped specimens. The stress-strain curves and resultant mechanical property values, particularly the elastic modulus and yield strength, which are the most essential parameters for determining the practical utility of the material, vary significantly according to the specimen dimensions (Figure 1e,f). Therefore, comparing the mechanical properties of tough hydrogels, even when dumbbell-shaped specimens are measured, remains a challenge. To the best of our knowledge, not enough studies have been conducted on the influence of dumbbell-shaped specimen shape and size on the measured mechanical properties of hydrogels.

**Figure 1 materials-16-00785-f001:**
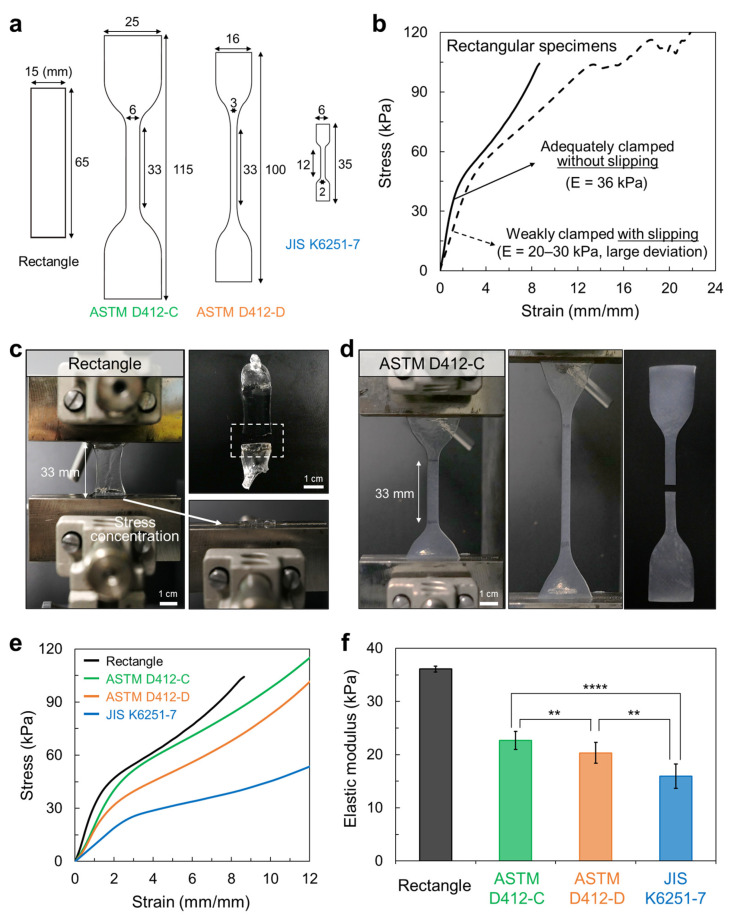
(**a**) Specimen dimensions of the rectangle, ASTM D412-C, ASTM D412-D, and JIS K6251-7. (**b**) Stress-strain curves for rectangular specimen under different clamping conditions. Photographs depicting (**c**) rectangular hydrogel and (**d**) ASTM D412-C hydrogel mounted in the testing machine and broken after the test. In the rectangular hydrogel, a stress concentration near the grip usually provokes an early fracture of the hydrogel (dashed box). (**e**) Stress-strain curve and (**f**) elastic moduli of the hydrogel with different shapes. ** *p* < 0.01; **** *p* < 0.0001.

### 3.2. Difference Caused by Gauge Length/Width of the Reduced Section of the Dumbbell-Shaped Specimen

To better understand the variation in the elastic modulus values according to the gauge length and width, we investigated the effect of the gauge length when maintaining other dimensions of the specimen at a constant value based on ASTM D412-C, which is a very frequently used dumbbell shape (Figure 2a). The tensile test results indicated that the overall slope of the stress-strain curve varied significantly according to the specimen gauge length (Figure 2b). Although almost no difference was observed in yield strength, the shorter gauge length resulted in a gentle slope; that is, the elastic modulus value decreased with a decrease in the gauge length (Figure 2c). This trend was consistently observed in other specimens, similar to ASTM D412-C, but with a much broader entire width. In this case, hydrogels with the same 6 mm narrow width of the reduced section, but with a 50 mm entire width (to accentuate the narrow reduced section with a wider clamping region), were prepared (Figure 2d). When the gauge length of the hydrogel was gradually reduced from 33 to 17 mm, the elastic modulus value gradually decreased (Figure 2e,f).

We hypothesized that there must be a more qualitative method to predict the measured elastic modulus values, depending on the dimensions of the dumbbell-shaped specimens. Thus, the modulus data depicted in Figure 2 were plotted using the mathematical expression of specimen dimensions, (*a* + *b*)/*ac*, where *a*, *b*, and *c* are the narrow width of the reduced section, entire width of the specimen, and gauge length, respectively (Figure 3). Notably, the data points in Figure 3 are linearly regressed. The difference between the two linear plots obtained from the specimens depicted in Figure 3 was expressed using the *b*/*a* ratio; the *b*/*a* values for each plot were 4.2 and 8.3 for the data points obtained from Figure 2a–c and Figure 2d–f, respectively. Under the same *b*/*a* ratio, a decrease in *c* leads to a reduced elastic modulus value. For a given (*a* + *b*)/*ac*, the specimen with a larger *b*/*a* ratio exhibited a higher modulus value. Taken together, we inferred that specimens with a longer (larger *c* and smaller (*a* + *b*)/*ac*) and narrower reduced section (smaller *a* and larger *b*/*a*) exhibited a larger elastic modulus value. In other words, when the significance of the reduced section is high, the elastic modulus is measured as a large value.

We further hypothesized that (*a* + *b*)/*ac* and *b*/*a* could be used as a parameter pair to estimate the experimental elastic modulus value of the hydrogel, where the widths and gauge lengths are associated with the accumulated stress in the reduced section. To test this idea, we prepared three dumbbell-shaped tough hydrogels with different dimensions but the same (*a* + *b*)/*ac* and *b*/*a* parameters (Figure 4a). All three specimens exhibited an almost similar pattern in the early region of the tensile stress-strain curve (Figure 4b) and the same elastic modulus value (Figure 4c). This result was consistent with our hypothesis that the experimental elastic modulus of tough hydrogels with different specimen dimensions is the same when they have the same (*a* + *b*)/*ac* and *b*/*a* parameters, despite having different specimen dimensions (Figure 4d).

In addition to the elastic modulus values that varied based on the specimen dimensions, the overall stress-strain curve pattern differed (Figure 5). The specimen with a narrower and longer reduced section relative to the entire width/length (i.e., larger *b*/*a* and smaller (*a* + *b*)/*ac*) exhibited a much steeper stress-strain curve (Figure 5a,b) and larger tangent modulus and tensile strength values at each strain point compared with those of other specimens (Figure 5d,e). The specimens with the same (*a* + *b*)/*ac* and *b*/*a* parameters exhibited almost overlapping stress-strain curves with a similar fracture strain, which led to similar strength values (Figure 5c,f). It is also notable that only tiny deviations were observed in the mechanical property values obtained from the different specimens with the same (*a* + *b*)/*ac* and *b*/*a* parameters (Figure 4c and Figure 5f). Owing to there being almost no deviations between the curves of each sample, particularly below a strain of 10, which is an elongation range for the practical use of soft materials, the well-designed dumbbell-shaped specimens with the same (*a* + *b*)/*ac* and *b*/*a* parameters that we propose are, in practice, favorable for evaluating and comparing the hydrogel mechanical properties.

### 3.3. Load Concentration in the Reduced Section Rather than the Clamping Region

In the previous sections, we demonstrated the importance of fabricating dumbbell-shaped specimens and the change in stress-strain curves based on the dimensions of dumbbell-shaped specimens. Thus, the significant differences caused by specimen shapes/sizes require attention to understand and accurately interpret the hydrogel mechanical properties. In this section, we revealed which specimen shape/size could be appropriate for evaluating soft/tough hydrogels by accessing the load-focusing region of the hydrogel specimen (Figure 6). To clearly visualize the change in the hydrogel during stretching, Alg/PAM hydrogels additionally crosslinked in FeCl_3_ solution were used as model hydrogels. Because the brown Fe-crosslinked Alg/PAM hydrogel displays a color change during stretching, distinguishing the region that gathers maximum load is easy.

A rectangular hydrogel distinctly exhibited a stress concentration near the grip (Figure 6a), similar to the previous result (Figure 1c); the bright yellow area near the grip indicates hydrogel deformation during polymer network stretching. The applied load should be evenly distributed over an area of 15 × 33 mm^2^; however, the load was locally concentrated in the area near the grip. Furthermore, the load was often considerably accumulated between the grips, which resulted in the abrupt failure of the hydrogel (Figure 6a arrows). Such an irregular stress distribution obscures the determination of material fracture elongation, as depicted in the stress-strain curves (Figure 6b). In the case of ASTM D412-C dumbbell-shaped hydrogels, a tall UTM machine with a high crosshead travel capacity is required; in particular, for highly stretchable double-network hydrogels that can stretch over a strain of 20–30, many testing machines may not be able to meet this capacity.

Therefore, a dumbbell-shaped specimen with a short gauge length can be a good alternative. Representatively, the specimen #5, with a reduced section with a 6 mm width and 17 mm length, exhibited highly consistent stress-strain curves with a stable stretching process (Figure 6c,d). The wide clamping region was stably fixed between the grips, and the applied load effectively gathered at the reduced section, which yielded highly similar stress-strain curves without deviations. Accordingly, while evaluating the mechanical performance of soft/tough hydrogels, the use of a rationally designed dumbbell-shaped specimen (such as a dumbbell with a wide clamping region and a properly narrow and long reduced section) is recommended, and accurate experimental results can be obtained and compared with previous results.

## 4. Conclusions and Outlook

Numerous efforts have led to many advances in mechanically robust tough hydrogels over the last two decades. Given such a development, comparing the mechanical properties of the reported tough hydrogels in a fair way is a necessity. We demonstrated the effect of the test specimen dimensions (width and gauge length) on the mechanical properties of tough hydrogels. In particular, the elastic moduli, a critical value for assessing hydrogel performance, of hydrogels with different shapes and sizes were systemically compared. A relationship between the specimen dimensions and measured elastic modulus value was derived, enabling the experimental elastic modulus value of the dumbbell-shaped tough hydrogel with different specimen dimensions to be estimated. Moreover, visual photographic materials exhibiting how the applied load transferred onto the hydrogel and gathered at the reduced section distinctly demonstrated the necessity for dumbbell-shaped specimens. This finding suggests that there is a need to start a discussion on how to evaluate the mechanical properties of soft/tough hydrogels and compare newly developed and previously developed hydrogels. Although further studies are mandated to firmly establish the conclusion, this finding provides baseline information regarding the testing method and the development of a standardized method for soft/tough hydrogels. Furthermore, since here we only used the tensile testing method, a comparison of experimental results obtained from different testing methods, including compression, indentation, and rheology tests, should be performed in order to reach a specific, true value of the material mechanical properties.

## Figures and Tables

**Figure 2 materials-16-00785-f002:**
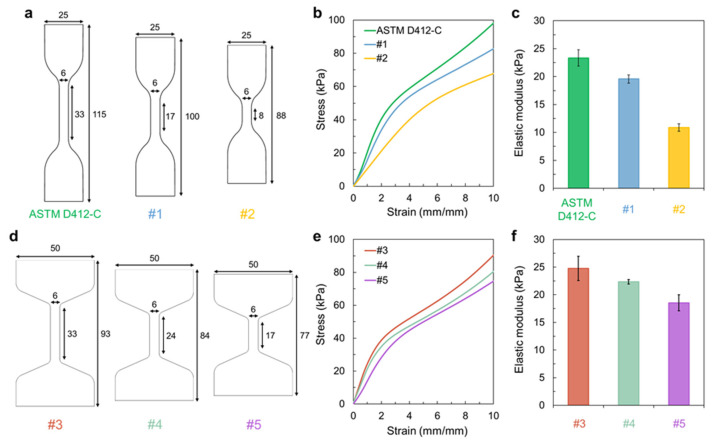
(**a**,**d**) Specimen dimensions of dumbbell shape with a different gauge length. (**b**,**e**) Stress-strain curve and (**c**,**f**) elastic modulus of hydrogels with different dimensions.

**Figure 3 materials-16-00785-f003:**
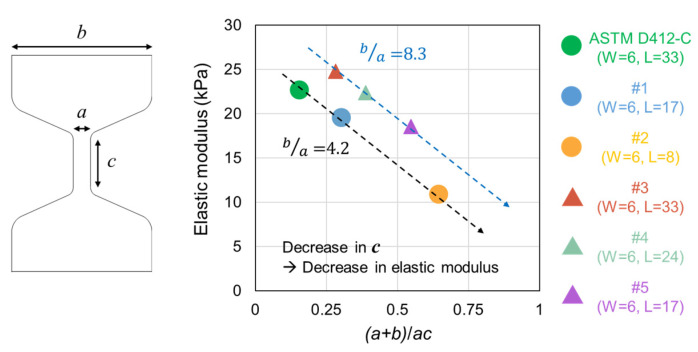
Elastic modulus value of hydrogels with a different gauge length and width, represented along (*a* + *b*)/*ac*.

**Figure 4 materials-16-00785-f004:**
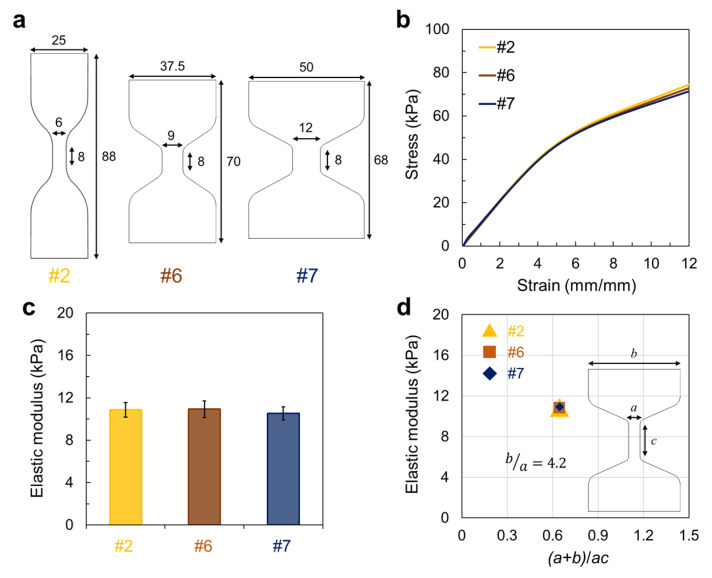
(**a**) Specimen dimensions of three dumbbell shapes with a fixed entire width/narrow width ratio of 4.17 and the same gauge length of 8 mm. (**b**) Stress-strain curve and (**c**) elastic modulus of each hydrogel. (**d**) Elastic modulus value of the hydrogel along (*a* + *b*)/*ac*.

**Figure 5 materials-16-00785-f005:**
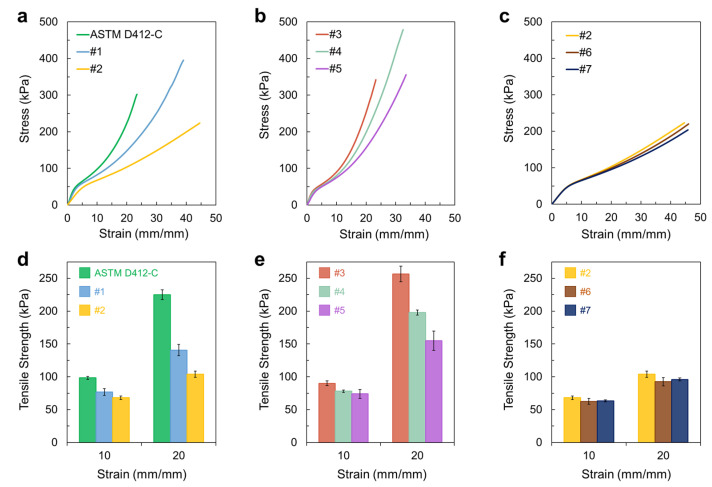
(**a**–**c**) Stress-strain curves of each hydrogel with a different specimen shape/size. (**d**–**f**) Tensile strength values at a strain of 10 or 20.

**Figure 6 materials-16-00785-f006:**
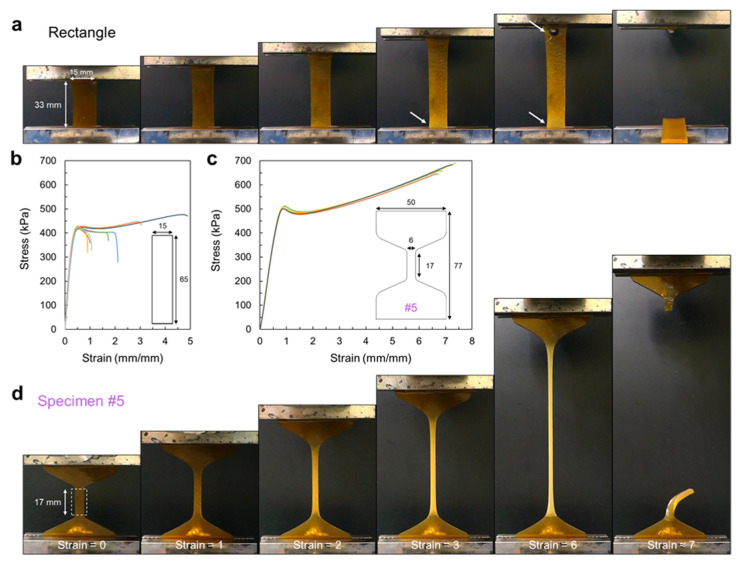
Photographs and stress-strain curves of Fe-crosslinked hydrogels with different shapes. (**a**,**d**) Photographs of a different stretching process for each hydrogel. Arrows in the photographs indicate the stress concentration near the grip resulting in the abrupt hydrogel rupture. (**b**,**c**) Stress-strain curves of each hydrogel. All five dumbbell-shaped specimens exhibited notably stable measurements, whereas rectangular specimens always exhibited different measurements.

**Table 1 materials-16-00785-t001:** Comparison of specimen shapes and sizes, and tensile testing speeds of representative tough hydrogels in the literature.

Hydrogels	Specimen Shapes	Specimen Size ^a^ Width × Length × Thickness [mm^3^]	Test Speed [mm min^−1^]	Strain Rate ^b^ [min^−1^]	Ref.
Alg/PAM	Rectangle	75 × 5 × 3	10	2.0	[2]
Alg/PAM	Rectangle	10 × N.M. × N.M.	60	N.M.	[6]
Alg/PAM	Dumbbell	2 × 12 × 2	100	8.3	[19]
Alg/PAM	Rectangle	5 × N.M. × 3	60	N.M.	[20]
Alg/PAM	Rectangle	5 × 30 × 3	60	N.M.	[21]
Alg/PAM	Rectangle	45 × 40 × 2	5	0.5	[22]
Alg/PAM	Dumbbell	2 × 35 × 1.8	100	2.9	[23]
Alg/PAM	Dumbbell	5 × 10 × 5	60	6	[24]
Agar/PAM	Dumbbell	4 × 30 × 1	50	1.7	[11]
Agar/PAM	Dumbbell	4 × 25 × 1	100	4.0	[12]
Chitosan/PAM	Rectangle	5 × N.M. × 2	50	N.M.	[13]
Chitosan/PAM	Rectangle	5 × N.M. × 1.5	10	N.M.	[14]
Chitosan/PAM	Rectangle	5 × 35 × 1.5	10	N.M.	[15]
Chitosan/PAM	Rectangle	N.M. × 10 × N.M.	N.M.	0.1	[16]
Cellulose/PAM	Dumbbell	4 × 16 × 4	40	2.5	[17]
PVA/PAM	Rectangle	75 × N.M. × 3	N.M.	2	[18]

^a^ N.M. indicates that the values were not mentioned in the paper. In the case of a dumbbell shape, the specimen size indicates the size of the reduced section (narrow width). Length is the gauge length, which is the distance between clamps in the rectangular shape or the length of the reduced section in the dumbbell shape. ^b^ The strain rate is the test speed (grip separation speed) divided by the gauge length. Alg: Alginate; PAM: polyacrylamide; PVA: poly(vinyl alcohol). Supplementary note: A non-contacting video extensometer can be used to precisely measure the distance change between benchmarks (gauge length) on the reduced section; however, some measurement challenges exist owing to the softness and wetting of the hydrogel. Moreover, a physically contacting extensometer can damage soft hydrogels. Therefore, extensometers have not yet been widely used in academic laboratories.

## Data Availability

The data supporting the findings of this study are available from the corresponding author.

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
