# Peer review of "Specimen Geometry Effect on Experimental Tensile Mechanical Properties of Tough Hydrogels"

_materials, 2023, doi:10.3390/ma16020785_

Round 1

Reviewer 1 Report

Dear Authors,

Your study could be interesting to the readers if it will be enriched with other investigations related to the chemical structure, morphological properties

The mechanical properties of a material depend on a multitude factors related to the physical properties of that material but also related to the chemical composition of it. Also, could be interesting to investigate the variations of the mechanical parameters of these hydrogels in conditions in which they are intented to be applied

Swelling behavior should be also invstigated and corelated to the other parameters of the hydrogem samples. The cross sectional morphology of the tensile fractured samples should be investigated also.

Please verify the unit measure in all graphics for the Stress Strain

Reviewer 2 Report

In this manuscript, the authors explored the effect of the specimen dimensions (width and gauge length) on the mechanical properties by using double-network hydrogels. They systematically explored the geometry (e.g., rectangle, dumbbell shape, gauge length, width, and clamping width) on the stress-strain curves and elastic modulus and derived a relationship between the specimen dimensions and measured elastic modulus values. Furthermore, the Fe3+-doped colorful hydrogel was employed to visualize the change in the hydrogel during stretching. This manuscript needs some revisions, and some suggestions were listed below for the improvement of this manuscript.

1.     The authors use (a+b)/ac to characterize the geometry, what’s the meaning of the expression?

2.     The hydrogels with the same (a+b)/ac demonstrate the same elastic modulus, but the stress-strain curves still different, how to understand this phenomenon?

3.     The geometry has influence on the mechanical property, which one is approaching the true values?  

Reviewer 3 Report

Look my comments

Round 2

Reviewer 3 Report

Weldone